# Diagnosis of Mucopolysaccharidoses and Mucolipidosis by Assaying Multiplex Enzymes and Glycosaminoglycans

**DOI:** 10.3390/diagnostics11081347

**Published:** 2021-07-27

**Authors:** Nivethitha Arunkumar, Dung Chi Vu, Shaukat Khan, Hironori Kobayashi, Thi Bich Ngoc Can, Tsubasa Oguni, Jun Watanabe, Misa Tanaka, Seiji Yamaguchi, Takeshi Taketani, Yasuhiko Ago, Hidenori Ohnishi, Sampurna Saikia, José V. Álvarez, Shunji Tomatsu

**Affiliations:** 1Nemours/Alfred I. duPont Hospital for Children, Wilmington, DE 19803, USA; niviarun@udel.edu (N.A.); Shaukat.Khan@nemours.org (S.K.); ssaikia@udel.edu (S.S.); Josevictor.Alvarezgonzalez@nemours.org (J.V.Á.); 2College of Health Sciences, University of Delaware, Newark, DE 19803, USA; 3Department of Endocrinology, Metabolism, and Genetics, Center for Rare Disease and Newborn Screening, National Children’s Hospital, Lathanh, Dongda, Hanoi 18/879, Vietnam; dungvu@nch.org.vn (D.C.V.); ngocctb@nch.org.vn (T.B.N.C.); 4Department of Pediatrics, Shimane University Faculty of Medicine, Izumo 693-8501, Japan; bakki@med.shimane-u.ac.jp (H.K.); seijiyam@med.shimane-u.ac.jp (S.Y.); ttaketani@med.shimane-u.ac.jp (T.T.); 5Clinical Laboratory Division, Shimane University Hospital, Izumo 693-8501, Japan; toguni@med.shimane-u.ac.jp; 6Shimadzu Corporation, Kyoto 604-8442, Japan; jun_wtnb@shimadzu.co.jp (J.W.); misa.tanaka@hotmail.co.jp (M.T.); 7Department of Pediatrics, Graduate School of Medicine, Gifu University, Gifu 501-1193, Japan; yago@gifu-u.ac.jp (Y.A.); ohnishih@gifu-u.ac.jp (H.O.)

**Keywords:** mucopolysaccharidoses, mucolipidosis, glycosaminoglycans, enzyme assay, newborn screening

## Abstract

Mucopolysaccharidoses (MPS) and mucolipidosis (ML II/III) are a group of lysosomal storage disorders (LSDs) that occur due to a dysfunction of the lysosomal hydrolases responsible for the catabolism of glycosaminoglycans (GAGs). However, ML is caused by a deficiency of the enzyme uridine-diphosphate N-acetylglucosamine:lysosomal-enzyme-N-acetylglucosamine-1-phosphotransferase (GlcNAc-1-phosphotransferase, EC2.7.8.17), which tags lysosomal enzymes with a mannose 6-phosphate (M6P) marker for transport to the lysosome. A timely diagnosis of MPS and ML can lead to appropriate therapeutic options for patients. To improve the accuracy of diagnosis for MPS and ML in a high-risk population, we propose a combination method based on known biomarkers, enzyme activities, and specific GAGs. We measured five lysosomal enzymes (α-L-iduronidase (MPS I), iduronate-2-sulfatase (MPS II), α-N-acetylglucosaminidase (MPS IIIB), N-acetylglucosamine-6-sulfatase (MPS IVA), and N-acetylglucosamine-4-sulfatase (MPS VI)) and five GAGs (two kinds of heparan sulfate (HS), dermatan sulfate (DS), and two kinds of keratan sulfate (KS)) in dried blood samples (DBS) to diagnose suspected MPS patients by five-plex enzyme and simultaneous five GAGs assays. We used liquid chromatography-tandem mass spectrometry (LC-MS/MS) for both assays. These combined assays were tested for 43 patients with suspected MPS and 103 normal control subjects. We diagnosed two MPS I, thirteen MPS II, one MPS IIIB, three MPS IVA, two MPS VI, and six ML patients with this combined method, where enzymes, GAGs, and clinical manifestations were compatible. The remaining 16 patients were not diagnosed with MPS or ML. The five-plex enzyme assay successfully identified MPS patients from controls. Patients with MPS I, MPS II, and MPS IIIB had significantly elevated HS and DS levels in DBS. Compared to age-matched controls, patients with ML and MPS had significantly elevated mono-sulfated KS and di-sulfated KS levels. The results indicated that the combination method could distinguish these affected patients with MPS or ML from healthy controls. Overall, this study has shown that this combined method is effective and can be implemented in larger populations, including newborn screening.

## 1. Introduction

Mucopolysaccharidoses (MPS) and mucolipidosis (ML II/III) are a group of inherited lysosomal storage disorders (LSDs) caused by the deficiency of specific lysosomal enzymes. These enzymes typically break down glycosaminoglycans (GAGs) in MPS, such as chondroitin sulfate (CS), dermatan sulfate (DS), heparan sulfate (HS), keratan sulfate (KS), and/or hyaluronic acid. However, ML is caused by a deficiency of the enzyme that tags lysosomal enzymes with a mannose 6-phosphate (M6P) marker for transport to the lysosome, resulting in a shortage of multiple lysosomal enzymes. With the deficiency of these enzymes, these GAGs accumulate in cells and lead to the dysfunction of multiple organs. Patients with MPS have specific manifestations, including corneal clouding, abnormal face and hair, skeletal dysplasia, respiratory problems with narrowing airway, frequent infectious diseases, hepatosplenomegaly, cardiac involvement, and/or cognitive impairment [1,2,3,4].

Although MPS is incurable, enzyme replacement therapy (ERT), hematopoietic stem cell transplantation (HSCT), and supportive therapies can manage symptoms in patients with MPS. Furthermore, infants are not typically born with unique clinical symptoms and develop characteristic symptoms later in their advancing period. However, the best prognosis for MPS patients depends on early diagnosis and early treatment [4,5,6,7,8], therefore, it is essential to quantify the accumulation of metabolic products such as GAGs for screening, diagnosis, and therapy monitoring. Each subtype of MPS can vary in elevated GAGs, confirming a diagnosis without genetic analysis. In advance of the appearance of clinical signs and symptoms, biomarkers (enzyme activity and GAG levels) indicate MPS present at birth and even in the fetus [9,10,11,12,13,14,15,16,17]. For this reason, it is critical to develop a newborn screening protocol and methodology of MPS that can be implemented in a statewide screening program. This newborn screening program can aid in the detection of disease before clinical signs and symptoms arise [18,19,20,21,22,23,24]. There are currently already over 20 states that have adopted newborn screening for MPS I [25,26,27,28]. These states use either fluorometric or mass spectrometric assays that measure the level of enzyme activities specific to each disease subtype [29,30].

Many groups, including our own, have reported a five-tier multiplex assay that can measure the enzyme activities for MPS I, II, IIIB, IVA, and VI simultaneously using tandem mass [31,32,33,34,35]. There are multiple methods for GAG analysis that have been summarized in detail by Kubaski et al. and Khan et al. [12,36]. These methods include the internal disaccharide method through enzymatic degradation developed by Oguma et al. to analyze GAGs from serum/plasma by utilizing chondroitinase B, keratanase II, and heparitinase enzymes, which digest polysaccharides to DS, KS, and HS disaccharides [37,38]; this method is the most common approach. Since GAG polymers are repeating units of disaccharides, enzymatic degradation results in a single disaccharide [24,39]. Auray-Blais et al. developed the methanolysis method followed by LC-MS/MS to identify GAGs (CS, DS, HS, and KS) from the urine of MPS patients [40]. The method has been adapted to identify and quantify GAGs from urine, cerebrospinal fluid (CSF), serum, and animal tissues [41,42,43,44,45]. The quantification of GAGs in LC-MS/MS by butanolysis derivatization, which was first developed by Trim et al., could significantly improve the quality of MS results, increasing the sensitivity for alkylated disaccharides by at least 70-fold [46,47]. The other method is the endogenous disaccharide method, where samples are prepared with hydrolases. The endogenous disaccharide method for GAG measurement shows the highest disease-to-non-disease discrimination [39,48]. Finally, the last method is the Sensi-Pro assay method developed by Lawrence et al. This method uses a combination of heparinases and the derivatization of the non-reducing ends of GAGs [48]. The terminal carbohydrate biomarkers can be distinguished and quantified by LC-MS/MS in samples including tissues, blood, and urine. A simple adaptation of the Sensi-Pro method is known as Sensi-Pro Lite, where the analytes are not derivatized and are analyzed using hydrophilic liquid chromatography (HPLC) columns.

In the present study, a simultaneous assay method was developed for five GAGs and was applied to newborn screening and for families with a history of LSD [10,11,13,24,49,50,51]. Moreover, the dried blood spots (DBS) from 43 subjects with clinically suspected MPS or ML were analyzed, and both GAG and enzyme activity levels for MPS I, II, IIIB, IVA, and VI were analyzed. This paper extends the previous research on the validation of liquid chromatography-tandem mass spectrometry (LC-MS/MS) for a five-plex assay for mucopolysaccharidoses [23,24,25,26,27,28,29,30,31].

## 2. Materials and Methods

### 2.1. DBS Samples

DBS samples from 43 patients with suspected MPS and 103 normal control subjects were collected at Gifu University, Shimane University, National Children’s Hospital, and Nemours/AIDHC. These samples were stored at −20 °C. The age of the patients ranged from 6 months to 11 years, in which thirteen individuals were females, and 30 were males. This study was approved by the IRB at the local institutes and Nemours/AIDHC (IRB#: 281498). The method was adapted from another study from our lab [33].

### 2.2. GAG Assay

Chondroitinase B, heparitinase, keratanase II, chondrosine (internal standard-IS), and the unsaturated disaccharides heparan sulfate, ΔDiHS-0S [2-acetamido-2-deoxy-4-O-(4-deoxy-a-ʟ-threo-hex-4-enopyranosyluronic acid)-ᴅ-glucose], ΔDi-HS-NS [2-deoxy-2-sulfamino-4-(4-deoxy-a-ʟ-threo-hex-4-enopyranosyluronic acid)-ᴅ-glucose, chondroitin ΔDi-4S [2-acetamido-2-deoxy-4-*O*-(4-deoxy-a-ʟ-threo-hex-4-enopyranosyluronic acid)-4-O-sulfo-ᴅ-glucose, keratan sulfate, mono-sulfated KS [Galß1-4GlcNAc(6S)], and di-sulfated KS [Gal(6S) Galß1-4GlcNAc(6S) were all provided by Seikagaku Co. (Tokyo, Japan). Stock solutions of the above disaccharides were used to make standard solutions by serial dilution consisting of 1000 ng/mL, 500 ng/mL, 250 ng/mL, 125 ng/mL, 62.5 ng/mL, 31.25 ng/mL, 15.625 ng/mL, and 7.8125 ng/mL of ∆DiHS-NS, ∆DiHS-0S, and ∆Di-4S and 10,000 ng/mL, 5000 ng/mL, 2500 ng/mL, 1250 ng/mL, 625 ng/mL, 312.5 ng/mL, 156.25 ng/mL, and 78.125 ng/mL mono-sulfated KS and di-sulfated KS. Then, 5 µg/mL IS was prepared in ddH_2_0 (Millipore Milli-Q Reference A+ System). Samples and standards were digested with chondroitinase B (0.5 mU/sample), heparitinase (1 mU/sample), and keratanase II (1 mU/sample). Acetonitrile optima, ammonium hydroxide, ammoniaque optima, ammonium acetate optima, bovine serum albumin, Corning Costar Assay plate 96-well, AcroPrep™ Advance 96-Well Filter Plates with Ultrafiltration Omega 10 K membrane filters (PALL Corporation, New York, NY, USA) were also used.

The LC-MS/MS apparatus consisted of a 1290 Infinity LC system with a 6460 triple quad mass spectrometer (Agilent Technologies, Palo Alto, CA, USA). Disaccharides were separated on a Hypercarb column (2.0 mm i.d., 50 mm length; 5 μm particles; Thermo Scientific, Waltham, MA, USA), thermostated at 60 °C. The method used was modified from the original one by Oguma et al. [38].

### 2.3. Multiplex Enzyme Assay

A disc (3.3 mm) was punched from each DBS sample using a DBS puncher. AcroPrep™ Advance 96-Well Filter Plates with Ultrafiltration Omega 10 K membrane filters (PALL Corporation, NY, New York, USA) were used for the enzyme assay. The reagents required to assay five enzymes (α-L-iduronidase (IDUA), iduronate sulfatase (IDS), α-N-acetylglucosaminidase (NAGLU), N-acetylglucosamine-6-sulfatase (GALNS), and N-acetylglucosamine-4-sulfatase (ARSB)) were purchased from PerkinElmer, Inc. A set of dried blood spots (DBSs) for quality control (QC) with high, middle, low, or base activity was obtained from PerkinElmer Inc. Tris Base and thermal adhesive sealing film were from Thermo Fisher Scientific, Ottawa, Ontario. Each DBS sample was placed into a 96-well plate with 30 µL assay cocktail containing known concentrations of the substrate: IDS to detect MPS II (0.5 mM), GALNS to detect MPS IVA (1 mM), and ARSB to detect MPS VI (1 mM).

Further, the DBS samples also contained internal standards for IDS (5 μM), GALNS (5 μM), and ARSB (5 μM) added to each well, and the whole plate was shaken at 37 °C for 16 h. The seal was removed and the mixture was combined with 100 uL of 50:50 methanol and extraction solution and mixed with a pipette. After that, quenching of the enzyme assay was followed by liquid-liquid extraction for purification. The mixture was then transferred to 96-well plates along with a 200 µL extraction solution and 200 µL ultrapure water followed by centrifugation for 5 min, and the top layers were transferred to the sampling plate. The sampling plate was evaporated at 40 °C for 5–10 min and shaken at room temp for 10 min after adding 100 µL of flow solvent to the plate. After this, the sample was measured by triple-quad MS/MS. The column analysis was performed using an LC-MS/MS system consisting of UHPLC with a triple quadrupole mass spectrometer (NexeraTM with LCMS-8050, Shimadzu Corporation, Kyoto, Japan). The mobile phases used were (A) 0.1% formic acid in water and (B) 0.1% formic acid in acetonitrile. LC-MS/MS with electrospray ionization was operated in multiple reaction-monitoring (MRM) mode.

### 2.4. Statistical Analysis

All statistical analyses and graphics were made through GraphPad Prism 8.0. Cut-off ratios for GAG values were calculated with the mean ± standard deviation of the control group. One-way ANOVA with Dunnett’s multiple comparison tests was performed to calculate the significance of results; a *p*-value of less than 0.05 was considered significant.

## 3. Results

The results from the combination assay indicated that the patients with MPS or ML could be identified from healthy controls. The age of the patients ranged from 6 months to 11.9 years, with an average age of 3.6 years. All of the patients had a suspected MPS, based upon clinical symptoms; therefore, this patient cohort is not indicative of the general population. From the enzyme assay and GAG assay of patients, we diagnosed 27 individuals as MPS or ML: two MPS I, 13 MPS II, one MPS IIIB, three MPS IVA, two MPS VI, and six ML patients. Of 43 patients, 27 had above the cut-off values determined from the control groups in at least one GAG (Table 1).

The cut-off values determined from the control groups were the mean + two standard deviations for each GAG. This control group was from previously collected and published data by Kubaski et al. [11,49]. The cut-off values for Di4S, DiHS-0S, and DiHS-NS were 42 ng/mL, 67 ng/mL, and 9 mg/mL, respectively. KS was age-dependent and, therefore, the cut-off values were calculated by the age group. The cut-off values for mono-sulfated KS can be found in Table 2, and the cut-off values for di-sulfated KS are in Table 3. Age was also considered when evaluating the di-sulfated KS/total KS percentage, as seen in Table 4.

Di4S, DiHS-0S, and DiHS-NS were significantly elevated in all patients diagnosed as MPS I (Figure 1, Figure 2 and Figure 3). Twelve of the 13 patients diagnosed with MPS II had an elevated level of Di4S, and all 13 patients had an elevated level of DiHS-0S. Twelve out of 13 MPS II patients had elevated levels of DiHS-NS. One patient diagnosed with MPS III had elevated DiHS-0S, DiHS-NS, and di-sulfated KS. All three patients diagnosed with MPS IVA had elevated di-sulfated KS and mono-sulfated KS. Di-sulfated Keratan Sulfate and Mono-sulfated keratan sulfate are both age dependant (Figure 4 and Figure 5) Both patients diagnosed with MPS VI had elevated Di4S, mono-sulfated KS, and di-sulfated KS (Figure 6 and Figure 7). Patients with MPS IIIB and MPS IVA did not have elvated levels of Di4S. MPS IVA and ML did not have elevated levels of DiHS-0S and DiHS-NS.

The enzymes assayed were IDUA, IDS, NAGLU, GALNS, and ARSB (Table 1). The patients identified with at least one significantly elevated GAG compared to that of the control were diagnosed with a subtype of MPS. The elevated GAG(s) in each patient were characteristics of the subtype of MPS. From Table 1, the enzyme assay for individuals was sensitive, and most of the patients only had one abnormally low quantification of enzyme activity. However, the exception was for patient 27, who had a low quantification of GALNS and ARSB. This patient was ultimately diagnosed with MPS VI because of the low enzyme activity and elevated levels of DS, which is not characteristic of MPS IVA. Six patients were diagnosed with ML based on the elevation of multiple enzymes. All six patients had elevated IDS, NAGLU, and ARSB. Four patients had elevated IDS, and one patient had elevated GALNS (Table 1).

All six patients diagnosed with ML had high concentrations of mono-sulfated KS and di-sulfated KS (Table 1, Figure 6 and Figure 7). One out of two MPS I patients had elevated di-sulfated KS and mono-sulfated KS concentrations. Three of the 13 patients diagnosed with MPS II had elevated mono-sulfated KS and di-sulfated KS, while 6 of the 13 MPS II patients only had elevated levels of di-sulfated KS. Furthermore, it was established that the values of mono-sulfated KS and di-sulfated were are age-dependent (Figure 4 and Figure 5).

## 4. Discussion

While the clinical symptoms of MPS disorders are not apparent at birth, biomarkers that can help diagnose MPS are present in newborns or even fetuses [49,52,53,54]. Measuring MPS enzyme activity and GAG levels is advantageous to screening and diagnosing MPS because significant differences in enzyme activity and biomarkers exist before the onset of clinical symptoms. It is also critical to predict the clinical severity in patients diagnosed before clinical symptoms arise so that appropriate treatment and management to the individual patient at an early stage can be provided. For this reason, it is crucial to create sensitive and reliable biomarkers that can correctly diagnose and provide a prognosis for individuals.

In this study, we screened 43 patients with suspected MPS derived from clinical symptoms and signs. We developed two-tier multiplex assays for enzyme activity and GAGs to diagnose five subtypes of MPS. The proposed two-tier assay is useful because it can decrease false positives during the diagnosis of MPS. Therefore, we have demonstrated that this screening system can help diagnose patients suspected of MPS based on clinical symptoms and that this assay could be applied to larger-scale populations, such as for newborn screening.

We have also found that the values of mono-sulfated KS and di-sulfated KS are age-dependent. Therefore, the cut-off value for the diagnosis of MPS needs age-matched controls. Depending on the age, there is a large variability of normal KS levels. This must be taken into account when developing cut-off values for KS. The elevation of mono-sulfated KS may help confirm the diagnosis of ML, MPS I, MPS II, MPS IVA, and MPS VI because mono-sulfated KS is significantly elevated in all these groups. Values of di-sulfated KS are also significantly elevated in ML, MPS I, MPS II, MPS IVA, and MPS VI.

GAG assays can help differentiate “true” enzyme deficiency and pseudodeficiency; therefore, a GAG assay is crucial to MPS newborn screening programs. A two-tier screening assay is necessary to increase sensitivity and specificity while decreasing false positives. The LC-MS/MS method for five enzymes and five GAGs can be applied in developing a more robust newborns screening program. This method uses DBS samples, which is the same collection process for samples for newborn screening protocols already in place. Most states in the screening for MPS I use enzyme activity by using microfluidics or MS/MS technology alone. Compared to a single assay with enzyme activity, the proposed method is more accurate and suited for the rigor of newborn screening programs (NBS). In the future, more pilot studies with a larger cohort of patients are necessary so that we can more accurately assess the sensitivity and specificity of the method described.

The first assay was a five-plex enzymes assay made through LC-MS/MS. Patients were suspected of having MPS biochemically if the enzyme levels were below the cut-off rates. Enzyme assays are the most common first-tier assay for LSDs since the deficient lysosomal enzyme specific to the disease enables the diagnosis to be more specific [16,20]. Enzyme assays can be done fluorometrically using an artificial substrate and a fluorescent tag such as 4-methylumbelliferone (4-MU) or through MS/MS. In this study, enzyme activities were measured with a LC-MS/MS method since LC-MS/MS applies to a multiplex assay [55]. If the patients had significantly decreased enzyme activity levels, a subtype of MPS was suspected. Patients who were significantly higher in multiple enzymes are diagnosed as ML due to the pathogenesis of ML and the clinical symptoms of LSDs [36,56].

Mucolipidosis is a lysosomal storage disorder caused by a deficiency of lysosomal enzyme-N-acetylglucosamine-1-phosphotransferase (GlcNAc-1-phosphotransferase) and results in the inability of lysosomal enzymes to be transported to the lysosome. This leads to the accumulation of GAGs, cholesterol, and phospholipids. Another LSD presents similarly to MPS, with clinical manifestations that include skeletal dysplasia, corneal clouding, and other clinical manifestations. Since the ML patients show clinical features similar to those with MPS, there needs to be an enzyme assay for GlcNAc-1-phosphotransferase or a genetic diagnosis. In many cases, this is difficult and expensive. Another method for diagnosis is analyzing the combined activities of other lysosomal enzyme activities. ML patients will have increased activity levels in other enzymes compared to normal controls, including arylsulfatase A and B and N-acetyl-β-galactosaminidase [36,57,58,59].

The multiplex GAG assay was completed in a similar way to the second-tier assay before a confirmatory diagnosis. After the initial enzyme assay, 27 patients had abnormal enzyme activity in one of 5 enzymes. As a second-tier, the GAG levels of 43 patients and 109 control patients were measured. Patients were diagnosed with a subtype of MPS or ML if they had abnormal enzyme activities followed by abnormal GAG values. According to the results, patients diagnosed with subtypes of MPS had increased specific GAG values concurrent with the current literature [60,61,62]. The cut-off values for diagnosis were if the GAG values were above 2 standard deviations from the mean of the control population.

KS is commonly elevated in MPS IVA as a primary storage material and is considered a biomarker for detecting this disease. However, it is also commonly elevated in other MPS types such as MPS I, II, IIIA, VI, and VII, which correlate with pathohistology [17,56,63]. KS is prevalent in the cornea, cartilage, and bones. In patients with MPS IVA, patients present with many skeletal abnormalities, such as short stature, kyphosis, and genu valgum [17,56,64]. Plasma KS is also increased in ML patients. Fujitsuka et al. have also demonstrated that mono-sulfated KS and di-sulfated KS were significantly elevated in patients with MPS II [65]. The fact that KS is elevated in other subtypes of MPS supports that KS is essential as a biomarker for MPS IV and other types of MPS and ML [17,56,66]. The severity of skeletal disease also correlated with KS elevation level in MPS I, IVA, and VII mice [63]. In the general population and MPS population, plasma KS varies with age—this appears in our results and other literature [56,67]. In healthy controls, the blood KS peaks between 0 and 2 years of age and gradually stabilizes [51,56]. However, blood KS is elevated in MPS and ML patients compared to that of age-matched controls. Skeletal dysplasia or excess of storage of GAGs may stimulate KS, leading to secondary elevation of KS in other types of MPS and ML. Another possible hypothesis for KS accumulation is that KS is co-deposited with other GAGs preventing KS from being metabolized or the secondary biochemical changes make KS more resistant to enzyme degradation. Such biochemical changes may include fucosylation, sialylation, and sulphations [17,56].

The results of our experiment have demonstrated that two-tier screening is essential in the newborn screening of MPS. Two-tier screening for MPS helps decrease false positives due to psuedodeficiencies and other abnormal increases in GAG values [12,68]. Other pilot studies for the NBS of MPS show similar low false-positive rates. In Taiwan, the false positive rate at the first-tier screening was 0.0037% for MPS I (294,196 DBS samples) and 0.107% for MPS II (153,032 DBS samples). After the repeat DBS and enzyme assay, the false-positive became 0.0014% for MPS I with four confirmed patients and 0.048% for MPS II with three patients [69]. In Italy (112,446 DBS samples), the false positive rate was 0.044% after the first-tier assay, and two patients were identified after the first-tier screening [12,70]. In Kentucky (55,161 DBS samples), the false positive rate at the first screening was 0.1% and 0.0018% after the second screening (one confirmed patient) [71]. In all these pilot studies, the enzyme assay was the first tier, and the GAG assay was the second tier. The endogenous disaccharide is the most effective method for second-tier GAG analysis for NBS [39,72]. In another recent study, LC-MS/MS was used to assay the enzyme activities of MPS-II, MPS-III, MPS-IVA, MPS-VI, and MPS-VII. This group concluded that NBS screening for MPS is feasible in an NBS laboratory because the number of patients positive after the initial screen was manageable to confirm [73]. Other possible additions to a two-tier screening pathway include a next generation sequencing (NGS) protocol to identify pathogenic mutations with a 100% specificity rate [74]. Overall, NBS is vital to ensuring MPS patients are treated quickly and efficiently.

A novel, rapid LC-MS/MS method was developed to simultaneously measure the activities of five lysosomal enzymes for newborn screening. This is one of several studies to implement a multiplex approach to diagnosing multiple subtypes of MPS with a two-tier MS/MS platform. There have been screening pathways developed for MPS II, MPS IVA, MPS VI, and MPS VII, but these assays do not conduct GAG assays to rule out false positives [35,61,62,64,75]. With the proposed two-tier method, a larger population of patients can be simultaneously screened for multiple subtypes of MPS with more sensitivity and specificity. The current study had several limitations related to adapting the screening to all types of MPS and ML. We screened a limited number of samples and diagnosed only five types of MPS, which are the most popular in the Asian population. Covering all types of MPS and ML is still an unmet challenge. A large pilot study is urgently required for newborn screening using the current strategy.

## 5. Conclusions

Enzyme assays are the gold standard for diagnosing MPS in patients, but a second-tier GAG assay must rule out false positives such as pseudodeficiency. The diagnosis of MPS patients should include assessing both the enzyme activity and the accumulation of metabolites such as GAGs. A combination assay can increase the sensitivity and specificity of a screening assay and allow for a more accurate patient diagnosis. This five-plex enzyme assay with GAG analysis is a potential method to broaden the range of newborn screening for MPS.

## Figures and Tables

**Figure 1 diagnostics-11-01347-f001:**
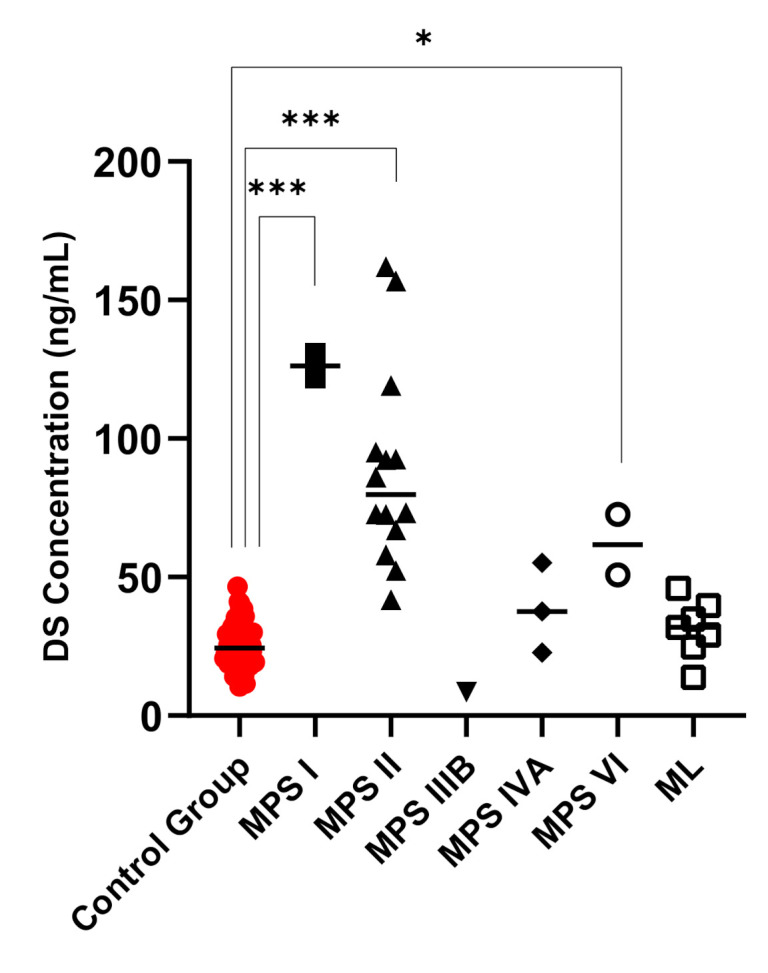
Dermatan sulfate (DS) values for control subjects and those diagnosed with MPS or ML. MPS I and MPS II had significantly increased DS values. Mean is shown with the horizontal bars. A *p*-value of <0.05 is denoted by * and a *p*-value of <0.001 is denoted by ***.

**Figure 2 diagnostics-11-01347-f002:**
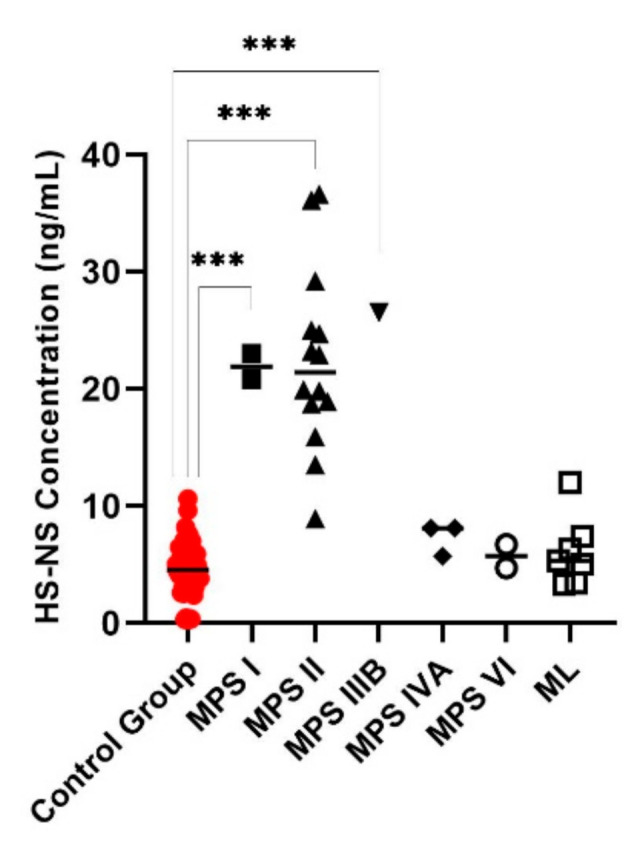
N-sulfated heparan sulfate (HS-NS) for control subjects and those diagnosed with MPS or ML. MPS I, II, and IIIB had significantly increased HS-NS values. Mean is shown with the horizontal bars. A *p*-value of <0.001 is denoted by ***.

**Figure 3 diagnostics-11-01347-f003:**
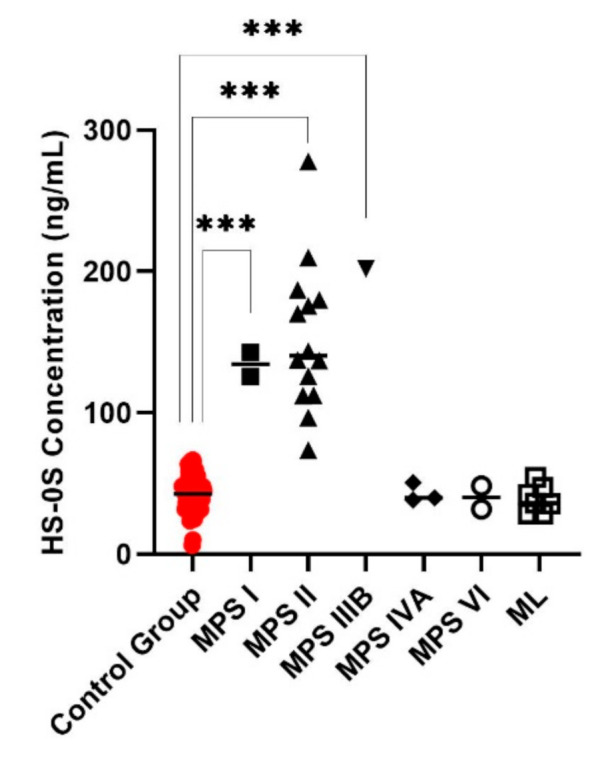
Non-sulfated heparan sulfate (HS-0S) for control subjects and those diagnosed with MPS or ML. The samples from MPS I, II, and IIIB had significantly increased HS-0S values. Mean is shown with the horizontal bars. A *p*-value of <0.001 is denoted by ***.

**Figure 4 diagnostics-11-01347-f004:**
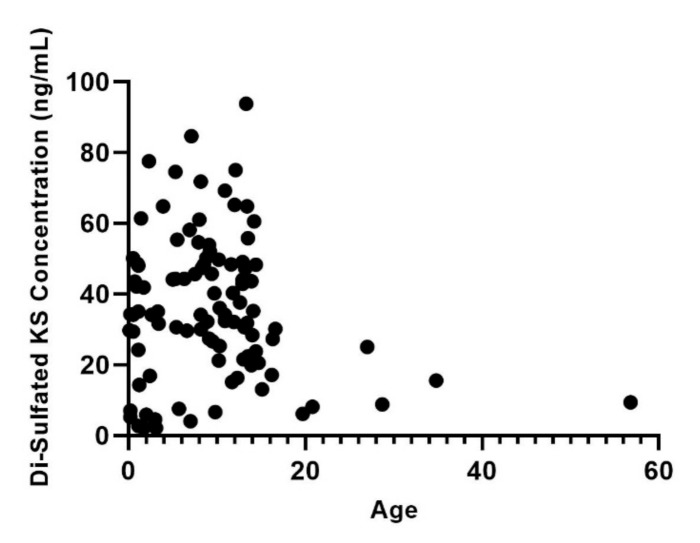
Di-sulfated keratan sulfate (KS) values for control subjects by age. All values are in ng/mL.

**Figure 5 diagnostics-11-01347-f005:**
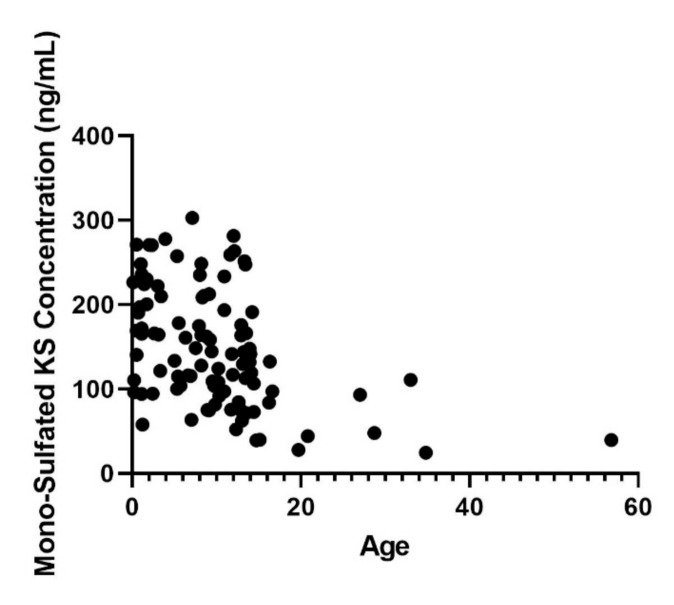
Mono-sulfated keratan sulfate (KS) values for control subjects by age. All values are in ng/mL.

**Figure 6 diagnostics-11-01347-f006:**
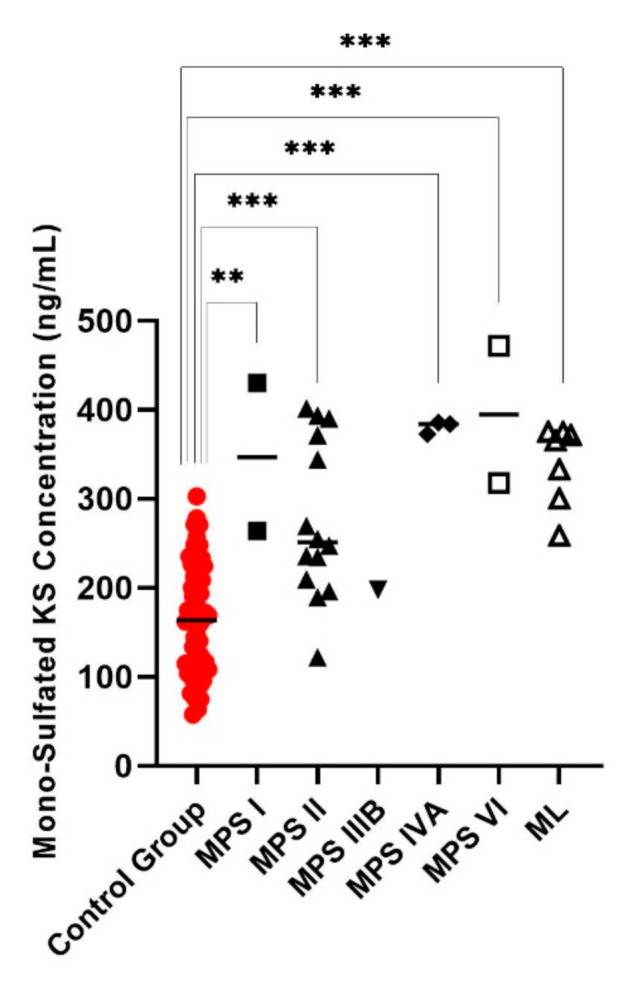
Mono-sulfated keratan sulfate (KS) values for control group and those diagnosed with MPS or ML. MPS IVA, VI, and ML had significantly increased mono-sulfated KS values. Mean is shown with the horizontal bars. A *p*-value of <0.01 is denoted by **, and a *p*-value of < 0.001 is denoted by ***.

**Figure 7 diagnostics-11-01347-f007:**
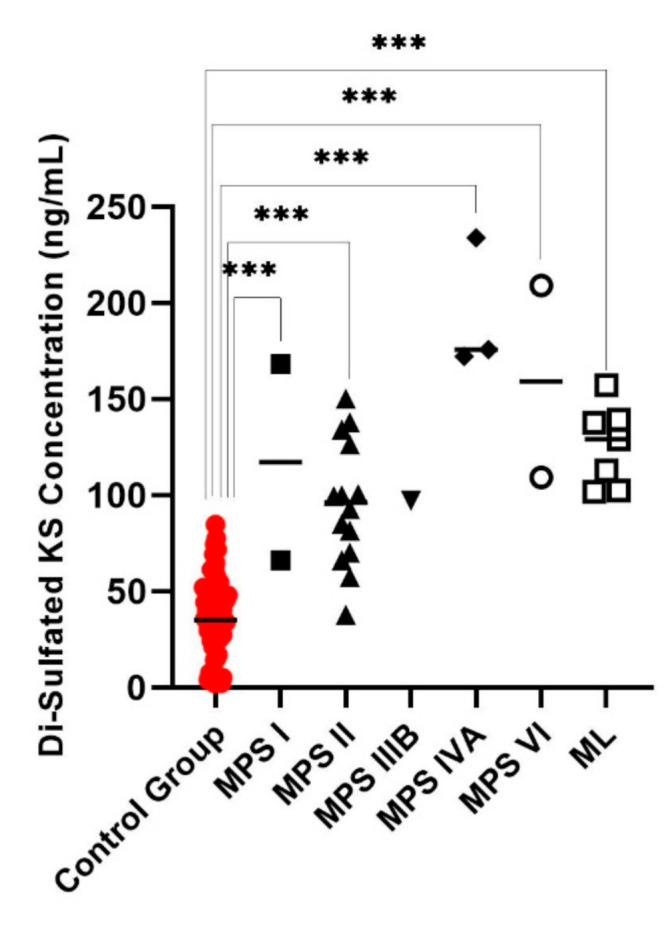
Di-sulfated keratan sulfate (KS) for control subjects and those diagnosed with MPS or ML. MPS IVA, VI, and ML had significantly increased di-sulfated KS values. Mean is shown with the horizontal bars. A *p*-value of <0.001 is denoted by ***.

**Table 1 diagnostics-11-01347-t001:** Diagnosed patients with GAG concentration and enzyme activity levels.

Case	Sex	Age	GAG Concentration (ng/mL)						Enzyme Activity ( μmol/hr/L)					Diagnosis
			DS	HS0S	HSNS	monoKS	diKS	Di/total KS %	MPS-I	MPS-II	MPS-IIIB	MPS-IVA	MPS-VI	
1	2	8.1	31.9	42.0	7.4	**374.6**	**112.9**	23.2	**9.34**	**103.74**	**21.81**	0.96	**6.70**	ML, High KS and enzyme activities
2	2	9.3	13.9	35.5	5.3	**371.5**	**157.2**	29.7	5.61	**132.16**	**24.79**	1.03	**5.77**	ML, High KS and enzyme activities
3	1	8.5	46.0	36.0	5.0	**300.4**	**102.3**	25.4	3.54	**198.48**	**35.03**	**1.81**	**5.52**	ML, High KS and enzyme activities
4	2	2.2	34.9	28.7	3.4	**365.8**	**139.3**	27.6	**11.22**	**141.07**	**34.59**	0.85	**5.45**	ML, High KS and enzyme activities
5	2	4.8	29.1	28.5	3.3	**375.2**	**137.6**	26.8	**14.05**	**103.87**	**34.48**	0.78	**4.46**	ML, High KS and enzyme activities
6	1	9.6	39.8	54.4	12.0	**333.1**	**129.1**	27.9	**14.46**	**112.96**	**26.32**	0.85	**6.99**	ML, High KS and enzyme activities
7	1	9.8	**121.8**	**142.9**	**23.0**	**430.1**	**168.3**	28.1	**0.44**	2.88	3.06	0.17	0.67	MPS I
8	1	0.9	**130.8**	**125.6**	**20.8**	263.9	66.2	20.1	**0.41**	5.44	2.57	0.43	1.36	MPS I
9	1	4	**58.1**	**125.9**	**18.7**	254.6	**100.6**	**28.3**	5.89	**0.41**	3.81	0.79	2.11	MPS II
10	1	2.5	**119.2**	**112.2**	**15.9**	247.3	70.0	22.1	5.91	**0.08**	4.40	0.66	2.50	MPS II
11	1	0.8	**72.7**	**175.6**	**23.2**	196.0	37.9	16.2	5.16	**0.06**	3.99	1.75	3.13	MPS II
12	1	5.1	41.9	**96.6**	**13.5**	235.6	**99.4**	29.7	3.72	**0.11**	3.70	0.29	1.08	MPS II
13	1	3.8	**73.2**	**73.7**	8.9	121.9	66.2	**35.2**	2.00	**0.05**	2.38	0.30	0.88	MPS II
14	1	2.5	**162.1**	**143.5**	**22.9**	234.3	**85.2**	26.7	3.51	**0.05**	5.12	0.35	1.06	MPS II
15	1	4.6	**72.8**	**170.2**	**24.7**	371.3	**126.3**	25.4	2.63	**0.05**	6.80	0.35	1.09	MPS II
16	1	5.5	**92.6**	**179.9**	**29.2**	344.0	**99.9**	22.5	3.28	**0.06**	2.22	0.29	0.90	MPS II
17	1	1.6	**95.1**	**277.8**	**36.6**	269.5	**92.8**	25.6	4.75	**0.08**	8.20	0.31	1.01	MPS II
18	1	5	**156.9**	**209.9**	**36.1**	393.7	**134.1**	25.4	3.23	**0.06**	4.07	0.34	1.19	MPS II
19	1	5	**92.3**	**112.4**	**19.8**	390.2	**137.9**	26.1	3.69	**0.06**	5.63	0.42	1.19	MPS II
20	1	1.1	**52.3**	**136.8**	**19.9**	189.3	57.3	23.2	8.71	**0.20**	5.57	0.82	1.84	MPS II
21	1	2.7	**67.1**	**186.8**	**25.0**	209.3	81.5	28.0	7.69	**0.19**	8.56	1.02	2.03	MPS II
22	1	3.9	8.6	**202.2**	**26.5**	197.7	**97.2**	**33.0**	4.42	6.51	**0.05**	0.47	1.59	MPS IIIB
23	2	5.9	37.7	50.6	8.1	**372.5**	**172.1**	**31.6**	2.35	6.47	2.76	**0.00**	1.01	MPS IVA
24	2	11.9	55.3	38.4	5.7	**385.6**	**175.7**	31.3	1.36	4.13	2.00	**0.00**	0.65	MPS IVA
25	1	6.4	22.8	39.9	8.1	**383.9**	**233.9**	**37.9**	2.30	3.74	2.26	**0.00**	1.17	MPS IVA
26	1	2.9	**72.6**	48.4	6.7	**317.8**	**109.3**	25.6	3.42	8.13	3.87	0.57	**0.00**	MPS VI
27	2	7.1	**50.8**	31.9	4.7	**472.0**	**209.1**	**30.7**	1.86	6.53	2.27	**0.13**	**0.00**	MPS VI

Note: Bolded GAG values signify abnormal values based on cut-off levels. Bolded enzyme activity levels show either significantly increased or decreased activity [29]. Dermatan sulfate (DS), heparan sulfate no sulfation (HS0S), heparan sulfate N (HSNS), mono sulfated keratan sulfate (monoKS), di sulfated keratan sulfate, keratan sulfate (KS), and mucolipidosis (ML).

**Table 2 diagnostics-11-01347-t002:** Cut-off values for mono-sulfated KS by age group [11].

Age (Years)	N	Mean (ng/mL)	SD (ng/mL)	Cut off (ng/mL)
0–4.9	26	185.6	62.8	311.34
5.0–10.0	27	147.5	54.5	256.63
10.0–15.0	33	136.3	62.3	260.84
15.0–20.0	5	76.3	42.8	161.79
over 20	6	60	34	127.99

Note: The number of patients in the control group, as well as the mean and standard deviation, are all included. KS; keratan sulfate, SD; standard deviation.

**Table 3 diagnostics-11-01347-t003:** Cut-off values for di-sulfated KS by age group [11].

Age (Years)	N	Mean (ng/mL)	SD (ng/mL)	Cut off (ng/mL)
0–4.9	26	28.8	18.9	66.6
5.0–10.0	27	41.5	17.8	77.1
10.0–15.0	33	39	16.1	71.3
15.0–20.0	5	18.7	10	38.7
over 20	6	20.2	18	56.2

Note: The number of patients in the control group, as well as the mean and standard deviation, are all included. KS; keratan sulfate, SD; standard deviation.

**Table 4 diagnostics-11-01347-t004:** Cut-off values for di-sulfated KS/total KS by age group [11].

Age (Years)	N	Mean (%)	SD (%)	Cut off (%)
0–4.9	27	13.6	7.4	28.29
5.0–10.0	23	21.8	4.5	30.81
10.0–15.0	34	22.3	4.58	31.46
15.0–20.0	5	20.1	3.76	27.57
over 20	6	23.9	9.74	43.37

Note: The number of patients in the control group, as well as the mean and standard deviation, are all included. KS; keratan sulfate, SD; standard deviation.

## Data Availability

All relevant data are within the manuscript. Any data interpretation is available on the request.

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
