# Peer review of "Diagnosis of Mucopolysaccharidoses and Mucolipidosis by Assaying Multiplex Enzymes and Glycosaminoglycans"

_diagnostics, 2021, doi:10.3390/diagnostics11081347_

Round 1

Reviewer 1 Report

  1. Please explain the relationship between the current study and the paper “Glycosaminoglycans as biomarkers for mucopolysaccharidoses and other disorders, diagnostics-1267898”.
  2. There were a lot of abbreviations in Table 1. Please specify them. Otherwise, it is hard to follow for readers. Why did authors not show the SD value for each data?
  3. Reviewer notice that the font size in Tables and Figures were pretty big. Please improve it.
  4. Authors should re-organize the section of Results in order to show the data clearly and logically. Moreover, authors described the results very simple, just gave some tables and figures. Please revise it.
  5. Table 2 to 4, the reference was added. These results were not the original data collected by authors?
  6. Please emphasize the limitation of your study.

Author Response

Responses to the comments;

  1. Please explain the relationship between the current study and the paper “Glycosaminoglycans as biomarkers for mucopolysaccharidoses and other disorders, diagnostics-1267898”.

Response: There is no relation between the two manuscripts regarding the materials since diagnostics-1267898 has analyzed serum samples from non-MPS and compared them with MPS samples. The current manuscript includes all DBS samples; therefore, the two articles are different in the samples.

  1. There were a lot of abbreviations in Table 1. Please specify them. Otherwise, it is hard to follow for readers. Why did authors not show the SD value for each data?

Answer: We specified all abbreviations in table 1 and other parts wherever appropriate (see the manuscript).

  1. Reviewer notice that the font size in Tables and Figures was pretty big. Please improve it.

Response; we have properly revised the size.

  1. Authors should re-organize the section of Results in order to show the data clearly and logically. Moreover, the authors described the results very simple, just gave some tables and figures. Please revise it.

Response: We have revised it according to the comments. We have explained tables and figures more in the text.

  1. Table 2 to 4, the reference was added. These results were not the original data collected by the authors?

Response: Yes. We have adapted the data from the original article. We have cited it properly.

  1. Please emphasize the limitation of your study.

Response: We have added the text in the Discussion as follows; “The current study has several limitations to adapt to all types of MPS and ML. We have screened the limited number of samples and diagnosed only five types of MPS, which are most popular in the Asian population. To cover all types of MPS and ML is still an unmet challenge. It is urgently required to have a large pilot study of newborn screening by using the current strategy.”

Reviewer 2 Report

This study evaluates the influence of GAG assessment in addition to the assessment of specific enzymes in the diagnosis of MPS and ML. It is a well written manuscript with clerly presented results. I have the following comments:

  1. In the introduction section please add a reference at the end of the first paragraph
  2.  The authors state that this combination of tests could be implemented towards larger populations. Would this be cost effective?

Author Response

Responses to the comments;

This study evaluates the influence of GAG assessment in addition to the assessment of specific enzymes in the diagnosis of MPS and ML. It is a well written manuscript with clerly presented results. I have the following comments:

  1. In the introduction section please add a reference at the end of the first paragraph

Response: We have added the following refs.

  • Tomatsu, S.; Averill, L.W.; Sawamoto, K.; Mackenzie, W.G.; Bober, M.B.; Pizarro, C.; Goff, C.J.; Xie, L.; Orii, T.; Theroux, M. Obstructive airway in Morquio A syndrome, the past, the present and the future. Mol. Genet. Metab. 2016, 117, 150–156.
  • Montaño, A.M.; Tomatsu, S.; Gottesman, G.S.; Smith, M.; Orii, T. International Morquio A Registry: Clinical manifestation and natural course of Morquio A disease. J. Inherit. Metab. Dis. 2007, 30, 165–174.
  • Tomatsu, S.; Montaño, A.M.; Oikawa, H.; Smith, M.; Barrera, L.; Chinen, Y.; Thacker, M.M.; Mackenzie, W.G.; Suzuki, Y.; Orii, T. Mucopolysaccharidosis type IVA (Morquio A disease): Clinical review and current treatment. Curr. Pharm. Biotechnol. 2011, 12, 931–945.
  • Tomatsu, S.; Mackenzie, W.G.; Theroux, M.C.; Mason, R.W.; Thacker, M.M.; Shaffer, T.H.; Montaño, A.M.; Rowan, D.; Sly, W.; Alméciga-Díaz, C.J.; et al. Current and emerging treatments and surgical interventions for Morquio A Syndrome: A review. Res. Rep. Endocr. Disord. 2012, 2012, 65–77.

  1.  The authors state that this combination of tests could be implemented towards larger populations. Would this be cost-effective?

Response: The proposed strategy utilizes the LC/MS/MS instrument for both GAGs and enzyme assays. Due to the sensitive and specific identification of GAGs and enzyme activities, LC/MS/MS is the most popular method. The current method uses multiplex enzymes and glycosaminoglycans; therefore, the cost is lower with larger samples. The kits for both assays are being made. Therefore, we can say this is a cost-effective method in a large study.

Round 2

Reviewer 1 Report

Thanks for your revised manuscript. I have no further comments.